# Visualising health risks with medical imaging for changing recipients' health behaviours and risk factors: Systematic review with meta-analysis

**Gareth J. Hollands** [1,2] *, **Juliet A. Usher-Smith** [3], **Rana Hasan** [3], **Florence Alexander** [3], **Natasha Clarke** [1], **Simon J. Griffin** [3,4]

**1** Behaviour and Health Research Unit, Department of Public Health and Primary Care, School of Clinical Medicine, University of Cambridge, Cambridge, United Kingdom, **2** EPPI-Centre, UCL Social Research Institute, University College London, London, United Kingdom, **3** Department of Public Health and Primary Care, School of Clinical Medicine, University of Cambridge, Cambridge, United Kingdom, **4** MRC Epidemiology Unit, Institute of Metabolic Science, School of Clinical Medicine, University of Cambridge, Cambridge, United Kingdom

* gareth.hollands@ucl.ac.uk

**Data Availability Statement:** The extracted data used in the review are available from the OSF

## Abstract

### Background

There is ongoing clinical and research interest in determining whether providing personalised risk information could motivate risk-reducing health behaviours. We aimed to assess the impact on behaviours and risk factors of feeding back to individuals' images of their bodies generated via medical imaging technologies in assessing their current disease status or risk.

### Methods and findings

A systematic review with meta-analysis was conducted using Cochrane methods. MED-LINE, Embase, PsycINFO, CINAHL, and the Cochrane Central Register of Controlled Trials (CENTRAL) were searched up to July 28, 2021, with backward and forward citation searches up to July 29, 2021. Eligible studies were randomised controlled trials including adults who underwent medical imaging procedures assessing current health status or risk of disease, for which personal risk may be reduced by modifying behaviour. Trials included an intervention group that received the imaging procedure plus feedback of visualised results and assessed subsequent risk-reducing health behaviour. We examined 12,620 abstracts and included 21 studies, involving 9,248 randomised participants. Studies reported on 10 risk-reducing behaviours, with most data for smoking (8 studies; $n$ = 4,308), medication use (6 studies; $n$ = 4,539), and physical activity (4 studies; $n$ = 1,877). Meta-analysis revealed beneficial effects of feedback of visualised medical imaging results on reduced smoking (risk ratio 1.11, 95% confidence interval [CI] 1.01 to 1.23, $p$ = 0.04), healthier diet (standardised mean difference [SMD] 0.30, 95% CI 0.11 to 0.50, $p$ = 0.003), increased physical activity (SMD 0.11, 95% CI 0.003 to 0.21, $p$ = 0.04), and increased oral hygiene behaviours

Project page (https://osf.io/bsf5e/; file name 'Final Data Extraction January 2022.xlsx').

**Funding:** The author(s) received no specific funding for this work.

**Competing interests:** The authors have declared that no competing interests exist.

**Abbreviations:** BMI, body mass index; CENTRAL, Cochrane Central Register of Controlled Trials; CI, confidence interval; CT, computed tomography; ICTRP, International Clinical Trials Registry Platform; LDL, low-density lipoprotein; MRI, magnetic resonance imaging; PRISMA, Preferred Reporting Items for Systematic Reviews and Meta-Analyses; SMD, standardised mean difference; UV, ultraviolet.

(SMD 0.35, 95% CI 0.13 to 0.57, $p = 0.002$). In addition, single studies reported increased skin self-examination and increased foot care. For other behavioural outcomes (medication use, sun protection, tanning booth use, and blood glucose testing) estimates favoured the intervention but were not statistically significant. Regarding secondary risk factor outcomes, there was clear evidence for reduced systolic blood pressure, waist circumference, and improved oral health, and some indication of reduced Framingham risk score. There was no evidence of any adverse effects, including anxiety, depression, or stress, although these were rarely assessed. A key limitation is that there were some concerns about risk of bias for all studies, with evidence for most outcomes being of low certainty. In particular, valid and precise measures of behaviour were rarely used, and there were few instances of pre-registered protocols and analysis plans, increasing the likelihood of selective outcome reporting.

## Conclusions

In this study, we observed that feedback of medical images to individuals has the potential to motivate risk-reducing behaviours and reduce risk factors. Should this promise be corroborated through further adequately powered trials that better mitigate against risk of bias, such interventions could usefully capitalise upon the widespread and growing use of medical imaging technologies in healthcare.

Author summary

### Why was this study done?

- There is ongoing clinical and research interest in determining whether providing personalised risk information could motivate risk-reducing health behaviours in recipients.

- One such intervention involves feeding back images generated via medical imaging technologies that assess an individual's current disease status or risk.

- A Cochrane review published in 2010 was unable to meaningfully reduce uncertainty concerning the intervention's effects due to the limited evidence available at that time.

### What did the researchers do and find?

- An updated systematic review with meta-analysis was conducted using Cochrane methods. Randomised controlled trials were identified that included an intervention group that received imaging plus feedback of visualised results and assessed subsequent behaviour.

- A total of 21 studies (involving >9,000 randomised participants) that reported on 10 risk-reducing behaviours were included.

- The primary outcome analysis revealed beneficial intervention effects on 6 behaviours including smoking, diet, and physical activity. There were concerns about risk of bias for all studies, and evidence for most outcomes was of low certainty.

**What do these findings mean?**

- Capitalising upon widespread and growing use of medical imaging technologies through feeding back medical images shows potential for motivating risk-reducing behaviours and reducing risk factors.

- More adequately powered trials that better mitigate against risk of bias would further reduce uncertainty around these effects.

## Introduction

Achieving and sustaining changes in health-related behaviours and associated risk factors is a vitally important challenge for population health, given that noncommunicable diseases account for an estimated 71% of all deaths worldwide each year [1]. Health behaviours that are in principle modifiable—such as smoking, alcohol use, and consumption of nutritionally poor diets—as well as linked risk factors—such as high systolic blood pressure, body mass index (BMI), fasting glucose, and low-density lipoprotein (LDL) cholesterol—are among the most significant risk factors globally for total disease burden [2].

The provision of risk information is pervasive within healthcare settings, in part due to expectations that it might motivate changes in recipients' health-related behaviours to modify their risks. As such, there is ongoing clinical and research interest in determining both the type of risk information and the means of delivery that could most effectively motivate such changes, particularly if this can capitalise on readily accessible practices and technologies. Providing personal risk information relating to, for example, genetic [3,4] or phenotypic risks of disease, including cardiovascular disease [5] and cancer [6], appears to have at best small effects on recipients' habitual health-related behaviours [7]. However, feedback of medical imaging results that directly reveal actual harm or impaired bodily function—for example, structural or functional bodily damage—attributable to a given behaviour, could plausibly offer a more potent approach [8,9]. Interventions of this type typically consist of an individual being shown medical images picturing his or her body together with some explanation, emphasising the implications of the results and how changes in behaviour can reduce risks to health.

Medical imaging allows access to personal information that was previously unavailable and invisible, enabling clinicians to produce assessments of existing bodily damage and disease progression and classify levels of future disease risk. Examples of such applications include computed tomography (CT) to assess arterial calcification, ultrasound to assess liver damage, and radiography to assess osteoporosis-related changes in bone density. Medical imaging has also been employed in nonclinical settings, such as for health promotion purposes within nominally healthy populations, an example being ultraviolet (UV) photography to assess sun-related skin damage. Imaging results usually require a degree of trained interpretation and as such need explanation for recipients to understand them. Feedback is often limited to oral and/or written descriptions or classifications, with variation in the extent to which individuals are actually shown scan images and results are explained [10]. Communication of the source images to individuals is not typically or systematically included within standard clinical practice—for example, not being mentioned in guidelines for carotid imaging procedures [11]—but is sometimes undertaken dependent on context and case, and may be preferred by patients

[12]. As accessibility and use of medical imaging technologies increases [13,14], so does their potential to motivate improvements in health behaviours and outcomes. As illustration of such widespread use, 44.9 million imaging tests were reported in England in the year 2018 to 2019, an increase of 9% on the previous year [15].

More generally, visual images are widely perceived to be an especially potent means of communication, reflected in two common idioms: "Seeing is believing" and "A picture is worth a thousand words". Medical imaging technologists and physicians tend to uphold such a view, in line with the societal discourse, that medical imaging scans reveal an objective truth and are synonymous with the actual body being imaged [16]. Within modern society's increasing saturation with visual images [17], the concept of seeing is commonly conflated with that of knowledge [18]. Psychological theory posits that, relative to abstract, conceptual information, processing of concrete stimuli such as imagery can engage automatic and emotionally evocative associations in memory and help form coherent links between the information presented and the implications for health and future mitigation of risks [19–22]. As such, visual imagery may be immediately comprehensible and impactful, reflected in an extensive and diverse body of research in public health and behavioural science highlighting the potency of aversive visual images for cognition and behaviour [23–27].

In this review, we aimed to assess the extent to which feeding back medical imaging results that enable individuals to visualise their own health risks, derived from a range of imaging technologies and health conditions, and within both clinical and nonclinical settings, can modify recipients' behaviours and risk factors. To our knowledge, this has not been comprehensively addressed by prior evidence syntheses, which have typically had a narrower focus. For example, these have assessed the impact of coronary artery calcium screening, but not specifically the role of visualised feedback, and have comprised solely or predominantly observational data, with little randomised evidence [28,29]. Systematic reviews of interventions to increase sun protection behaviours have also been conducted but have typically assessed a wide array of interventions, as opposed to those centred on medical imaging [30,31]. Most pertinently, a Cochrane review of visual feedback of medical images published in 2010 [32] was unable to reduce uncertainty on this topic due to the limited evidence available at that time, but this has increased greatly since. This updated version of that Cochrane review aims to inform considerations as to whether the widespread use of medical imaging offers a largely untapped opportunity for improving health-related behaviours and reducing risk factors.

## Methods

This is an updated version of a registered Cochrane review (CD007434) from 2010 [32], adding 12 studies (of 7,877 participants) to the previous 9 studies (of 1,371 participants). Changes to the protocol since the 2010 review reflect updated Cochrane guidance on conduct of systematic reviews, including completing a GRADE assessment for each primary outcome [33], and assessment of study-level risk of bias using the revised RoB 2 tool [34] (see Appendix A in S1 Text). This study is reported as per the Preferred Reporting Items for Systematic Reviews and Meta-Analyses (PRISMA) guideline (S1 Checklist). Ethical approval was not required for this study.

### Data sources

We initially searched MEDLINE, Embase, PsycINFO, CINAHL, and the Cochrane Central Register of Controlled Trials (CENTRAL) up to December 17, 2019. These searches were subsequently updated to July 28, 2021, with backward and forward citation searches (in Google

Scholar) conducted to July 29, 2021. Coverage of grey literature was via CENTRAL—which includes records from key clinical trials registries (ClinicalTrials.gov; International Clinical Trials Registry Platform (ICTRP))—as well as via Embase and Google Scholar searches that include conference proceedings and preprints. The search required an abstract in English but was otherwise not restricted by language. For any articles identified as potentially eligible based on their title–abstract record, Google Translate was used for determining potential eligibility of the full text in the first instance, followed by translation as necessary. RH and FA ran the initial database searches, with GJH conducting the updated searches. Appendix B in S1 Text details the search strategy for all databases.

## Inclusion and exclusion criteria

**Study design.**   Eligible studies were randomised controlled trials. Cluster randomised controlled trials, with randomisation by site, were eligible providing the study included at least 2 intervention sites and 2 control sites.

**Population.**   Eligible participants were adults ($\geq$18 years) who underwent medical imaging procedures assessing current health status or risk of diseases (such as cancers or cardiovascular disease), for which personal risk may be reduced by modifying behaviour.

**Intervention.**   Eligible studies required an intervention group that received the imaging procedure plus feedback of visualised results, with this being the sole or principal intervention being assessed. Medical imaging was defined in relation to the MeSH definition of diagnostic imaging [35] and included magnetic resonance imaging (MRI), CT, radiography, ultrasonography, and UV photography. Feedback meant the individual being shown generated images of or representing his or her body (e.g., an ultrasound scan of an arterial plaque) along with some explanation of what the image portrays and its implications for health and/or behaviour (e.g., highlighting the influence of smoking on arterial health). We excluded interventions which fed back nonpersonalised, generic images not derived from medical imaging or used images of other people as the focus of communication; studies that used hypothetical scenarios in which individuals imagined receiving risk information; studies of multicomponent interventions that included additional substantive components unrelated to the communicated feedback whose effect could not be disentangled; and those in which the visual images purposefully altered the morphological properties of the body, e.g., by disfiguring or ageing.

**Comparison.**   Eligible comparison groups were those that did not undergo a medical imaging procedure but were provided with personalised risk information derived from a nonmedical imaging method (e.g., cholesterol test); those that underwent medical imaging and were provided with personalised risk information (e.g., verbal or numeric information) derived from the procedure but without being provided with any visual feedback of those medical images; or those that were provided with no personalised risk information at all (e.g., provided with only generic health risk information), whether or not they underwent a medical imaging procedure.

**Outcome.**   Primary outcomes were health-related behaviours that if changed could reduce health risks, including smoking, skin self-examination, physical activity, dietary behaviour, medication use, sun protection behaviours, oral hygiene behaviours, tanning booth use, foot care, and blood glucose testing. Eligible trials had to assess at least one of these primary outcomes. Secondary outcomes were risk factors and indices, including Framingham risk score, systolic and diastolic blood pressure, LDL cholesterol level, fasting glucose, glycated hemoglobin, BMI, waist circumference, skin darkening due to UV exposure, and oral health, as well as adverse events and harms, such as anxiety, depression, and stress.

## Screening, data extraction, and synthesis

Two authors prescreened all search results (titles and abstracts) against the inclusion criteria. For the initial searches, this process was conducted by GJH and RH or FA; for the updated searches, this was conducted by GJH and NC. Studies selected by either or both authors were subjected to full-text assessment. At least 2 authors independently assessed the selected full-text articles for inclusion (GJH and RH or FA for the initial searches; GJH and NC for the updated searches), with additional authors (JUS and SJG) acting as arbiter if required. Two authors (GJH and RH or FA) independently extracted key information including study characteristics and outcome data. Any disagreements were resolved by discussion with additional authors as necessary (JUS and SJG). Extracted outcome data were checked against the full-text articles by an additional author (NC). One author (GJH) entered the outcome data for analysis into Review Manager software, with data entry checked by a second author (NC). We contacted study authors to obtain missing primary outcome data.

Studies were analysed by each type of primary and secondary outcome, summarising effect sizes using forest plots. Effect sizes for dichotomous data were expressed as risk ratios and for continuous outcomes were standardised mean differences (SMDs) or mean differences if the same measures were used. When data for the same outcome were provided as dichotomous data by some studies and as continuous data by others, we first converted the log odds ratios for the dichotomous outcome data to SMDs using methods outlined by Chinn and in the Cochrane Handbook [36,37]. Second, we then combined the SMDs across all studies for that outcome using (generic) inverse variance–weighted meta-analysis, with each study only providing a single effect estimate to any given meta-analysis. For all meta-analyses, we obtained pooled effect sizes with 95% confidence intervals (CIs) using a random effects model. We tested for heterogeneity using the chi-squared test, applying a threshold of $p = 0.10$ due to the likelihood of this being underestimated in small samples [38], and quantified it using the $I^2$ statistic, with a value of 50% or greater considered to represent substantial heterogeneity (albeit noting that with limited numbers of studies CIs around this value will be wide). If the specified thresholds were met for either chi-squared or $I^2$ values or both, heterogeneity was further explored. In exploring possible sources of heterogeneity, we focused on 4 key characteristics: (i) outcome time point; (ii) outcome measure; (iii) health condition being imaged; and (iv) nature of the control group (namely whether control participants also underwent the medical imaging procedure, but without receiving visual feedback). Due to insufficient studies to justify meta-regression or subgroup analyses, this was investigated by conducting sensitivity analyses for each of these characteristics when it was possible to do so. We examined the effect on heterogeneity and the effect estimate of removing those studies that differed from the modal characteristic. This process was necessarily dependent on there being at least 3 studies, with overlap in their characteristics, within the respective meta-analysis; if there were only 2 studies, differences in their characteristics were noted.

If data were presented for more than one time point after the intervention, we used outcomes for the longest follow-up available. In any given behavioural domain, if multiple outcomes were reported, we used the outcome judged to be most important for reducing the specified health risk. If multiple indices or measures of a given outcome were reported, we used the most stringent and valid that was available (e.g., an objective measure such as biochemically validated smoking cessation). Additional detail on the process of selecting outcome data for analysis is provided in Appendix C in S1 Text. We used final values rather than changes from baseline wherever possible, avoiding combining these in analyses of continuous data unless they involved mean differences on the same scale. We analysed available data as reported according to participants' randomised groups. We did not attempt to impute missing

data (such as by assuming that participants with missing outcomes were engaging in a risk increasing behaviour).

## Assessments of risk of bias and certainty of evidence

Two review authors (GJH and RH, FA, or NC) independently assessed risk of bias in accordance with the RoB 2 tool [34], with an additional author acting as arbiter if necessary. We derived an overall summary judgement (Low; Some concerns; High) for each study. The summary risk of bias contributed to a GRADE assessment [33] of the certainty of evidence pertaining to the effect estimates for each primary outcome, assessing the likelihood that the true intervention effect will not be substantially different from what the research found. This is based on the design of the underlying studies—with randomised controlled trials initially treated as of high certainty—and on additional factors that decrease or increase certainty. GRADE criteria for downgrading certainty of evidence encompass study limitations, inconsistency, imprecision, indirectness, and publication bias. The evidence can be downgraded by one or 2 levels for each criterion, depending on whether concerns are, respectively, serious or very serious. One of four overall certainty ratings can ultimately be applied for each outcome. These certainty ratings are as follows: high certainty, whereby the current evidence provides a very good indication of the likely effect, and the likelihood that the true effect will be substantially different is low; moderate certainty—it provides a good indication of the likely effect and the likelihood that the true effect will be substantially different is moderate; low certainty—it provides some indication of the likely effect and the likelihood that the true effect will be substantially different is high; and, finally, very low certainty, whereby the current evidence does not provide a reliable indication of the likely effect and the likelihood that the true effect will be substantially different is very high.

## Results

We screened 12,620 title-abstract records, and 21 studies (including 9,248 participants) met the inclusion criteria. Fig 1 outlines the search and screening process, and Table 1 gives details of the included studies.

Most (14/21) studies were undertaken in inpatient or outpatient healthcare or clinical research settings [39–52], with the remainder conducted in nonclinical research or community settings [53–59]. In terms of imaging technologies used and the health conditions being investigated, nine studies imaged cardiovascular risk, specifically arterial, venous or cardiac health, using either ultrasound [39,40,46,50,52] or CT [44,45,48,51]; six studies used UV photography to image UV exposure–related skin damage [53–58]; three studies used oral photography (intraoral or quantitative light fluorescence photography) to image gingivitis, dental plaque, or tooth decay within dental consultations [41–43]; one used whole body photography to aid in monitoring new and changing skin moles [47]; one used photography to image skin health related to fruit and vegetable consumption [59]; and one used fundus photography to image retinal health linked to diabetes [49]. A total of 11 studies were conducted in the USA [40,44,47,48,51,53–58], three in the UK [43,52,59], two in Portugal [41,42], and one each in Australia [49], Sweden [46], Denmark [45], Switzerland [50], and the Seychelles [39]. Mean participant ages ranged from 20 to 63 years, and the proportion that were female ranged between 0% and 100%. The duration of the feedback intervention, if reported, was an hour or less and typically occurred at a single time point; in one case, participants were resent the feedback information and a phone call checked comprehension [46].

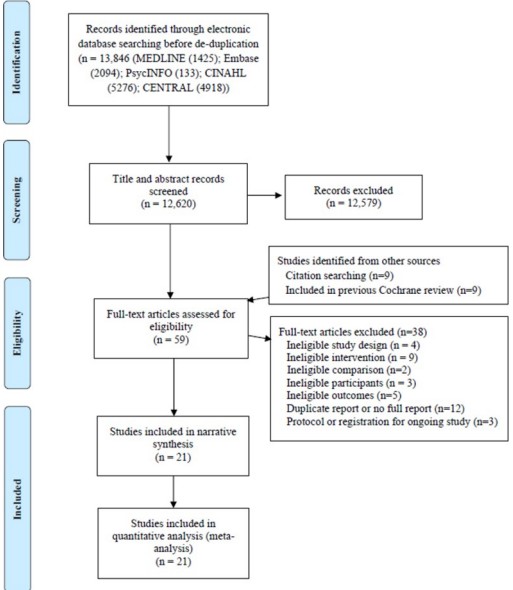

**Fig 1. PRISMA flow diagram of identification and selection of studies.** CENTRAL, Cochrane Central Register of Controlled Trials; PRISMA, Preferred Reporting Items for Systematic Reviews and Meta-Analyses.

## Primary outcome analysis

Separate forest plots display the results for all 10 primary behavioural outcomes, grouped into those outcomes for which there are dichotomous outcome data only (Fig 2: smoking and skin self-examination), those with combined dichotomous and continuous outcome data (Fig 3: physical activity, diet, and medication use), and those with continuous outcome data only (Fig 4: sun protection, oral hygiene behaviours, tanning booth use, foot care, and blood glucose testing).

**Smoking.** Eight studies [39,43,45,46,48,50–52] assessed current smoking status, or cessation in known smokers, up to 48 months postintervention, the modal follow-up being 12 months. Seven studies used either ultrasound or CT to visualise cardiovascular health [39,45,46,48,50–52], while one study used quantitative light fluorescence photography to visualise oral health [43]. Pooled analysis ($n$ = 4,308) showed that imaging feedback led to an increase in the proportion of participants not smoking (risk ratio 1.11, 95% CI 1.01 to 1.23, $p$ = 0.04, $I^2$ = 44% ($I^2$ CI 0% to 74%); Fig 2). Evidence for this outcome was assessed to be of moderate certainty using the GRADE approach.

**Skin self-examination.** A single study [47] ($n$ = 85 analysed) showed that feedback of whole body photography to aid in monitoring new and changing moles increased skin self-examination behaviour at four months (risk ratio 2.71, 95% CI 1.44 to 5.11, $p$ = 0.002; Fig 2). Evidence for this outcome was assessed to be of low certainty using the GRADE approach.

**Physical activity.** Four studies [45,48,50,51] assessed self-reported physical activity, with follow-up of up to 48 months and the modal time point being 12 months. All evaluated either ultrasound or CT to visualise cardiovascular health. Pooled analysis ($n$ = 1,877) showed that imaging feedback led to increased physical activity (SMD 0.11, 95% CI 0.00 to 0.21, $p$ = 0.04, $I^2$ = 0% ($I^2$ CI 0% to 68%); Fig 3); presenting this effect size estimate to three decimal places shows that the CI does not include zero (SMD 0.105, 95% CI 0.003 to 0.207). Evidence for this outcome was assessed to be of low certainty using the GRADE approach.

**Table 1. Characteristics of included randomised controlled trials.**

| Study | Country | Setting and participants (number randomised) | Intervention including type of medical imaging and related health condition | Comparison | Primary outcome(s) selected for review; timing of assessment / secondary outcome(s) selected for review; timing of assessment |
|---|---|---|---|---|---|
| Araújo (2016) [41] | Portugal | Patients with gingivitis receiving supportive periodontal therapy at dental clinic ($n = 80$) | Intraoral camera photography of gingivitis and dental plaque used to show and discuss with patient (within dental hygiene consultation) | As intervention but without use of intraoral camera photography | Oral hygiene behaviours—Brushing and flossing frequency (self-report); 4 months/Oral health; 4 months |
| Araújo (2019) [42] | Portugal | Patients with gingivitis receiving dental hygiene consultation at dental clinic ($n = 203$) | Intraoral camera photography of gingivitis and dental plaque used to show and discuss with patient (within dental hygiene consultation) | As intervention but without use of intraoral camera photography | Oral hygiene behaviours—Brushing and flossing frequency (self-report); 8 months/Oral health; 8 months |
| Athar (2018) [40] | USA | Patients hospitalised with acute decompensated heart failure ($n = 97$) | Ultrasound imaging of inferior vena cava to show degree of distention and excess fluid linked to heart failure and cardiovascular disease (within educational heart failure consultation) | Ultrasound imaging as per intervention but without viewing images and with accompanying generic information only, i.e., no imaging-related information | Medication use—Use of heart failure medication regimen as prescribed (self-report), Healthier diet—Use of a low-salt diet (self-report); 1 month |
| Bovet (2002) [39] | Seychelles | General population smokers drawn from Seychelles Heart Study II to attend study clinic ($n = 155$) | Ultrasound imaging of arterial health with photographs given if atherosclerotic plaque identified (within smoking cessation counselling intervention) | As intervention but no ultrasound imaging; smoking cessation counselling only | Smoking—Smoking cessation as 7-day abstinence (self-report); 6 months |
| Gibbons (2005); Study 2 [53] | USA | Laboratory study with university students ($n = 134$) | UV photography for sun damage, with skin damage explained in relation to UV photos (plus regular photo and oral presentation about the role of UV radiation exposure in cancer and photoaging) | As intervention but no UV photography (plus regular photo taken, and oral presentation about the role of UV radiation exposure in cancer and photoaging) | Tanning booth use—Frequency of tanning booth use (self-report); 1 month |
| Harris (2020) [43] | UK | Patients at medium-high risk of poor oral health receiving dental practice consultation ($n = 412$) | Quantitative light fluorescence photography of mouth for tooth decay and dental plaque (in addition to standard verbal advice within dental consultation) | As intervention but no photography, standard verbal advice only | Smoking—Current smoking status (self-report), Oral hygiene behaviours—Duration of brushing teeth (self-report), Healthier diet—Frequency of eating sugary foods (self-report); 12 months/Oral health; 12 months. Primary outcome data requested and received for smoking, oral hygiene, and healthier diet (in NIHR HS&DR report "Presenting patients with information on their oral health risk: the PREFER three-arm RCT and ethnography") |
| Lederman (2007) [44] | USA | General population postmenopausal women without coronary artery disease history receiving conventional cardiac risk screening ($n = 56$) | CT scan and feedback of images including categorisation into 1 of 4 categories of risk for coronary artery disease (plus conventional screening assessment for cardiac risk and counselling session based on results of conventional screening) | As intervention but no CT scan conducted (conventional screening assessment for cardiac risk and counselling session based on results of conventional screening) | Medication use—Increase in cholesterol medication use (self-report), Healthier diet—Decrease in fat intake (self-report); 12 months/Systolic blood pressure, diastolic blood pressure, LDL cholesterol, glycated hemoglobin, BMI; 12 months |

(*Continued*)

**Table 1.** (Continued)

| Study | Country | Setting and participants (number randomised) | Intervention including type of medical imaging and related health condition | Comparison | Primary outcome(s) selected for review; timing of assessment / secondary outcome(s) selected for review; timing of assessment |
|---|---|---|---|---|---|
| Mahler (2003); Study 2 [54] | USA | Community study at beaches with beachgoers (*n* = 76) | UV photography for sun damage, with skin damage explained in relation to UV photos (and given photoaging information brochure or not (in factorial design)) | As intervention but no UV photography (and given photoaging information brochure or not (in factorial design)) | Sun protection—Sun protection index for behaviours during intentional exposure (self-report); 1 month |
| Mahler (2006) [55] | USA | Community study at beaches with beachgoers (*n* = 244) | UV photography for sun damage, with skin damage explained in relation to UV photos | No UV photography, only questionnaires completed | Sun protection—Sun protection behaviours index (self-report); 2 months/Skin darkening; 2 months |
| Mahler (2007) [56] | USA | Laboratory study with university students (*n* = 133) | UV photography for sun damage, with skin damage explained in relation to UV photos (plus regular photo and photoaging information video shown or not (in factorial design)) | As intervention but no UV photography (plus regular photo taken, and photoaging information video shown or not (in factorial design)) | Sun protection—Sun protection behaviours index (self-report); 12 months. Primary outcome data requested and received for sun protection |
| Mahler (2013) [57] | USA | Laboratory study with university students (*n* = 442) | UV photography for sun damage, with skin damage explained in relation to UV photos (plus regular photo and photoaging information video shown or not (in factorial design)) | As intervention but no UV photography (plus regular photo taken, and photoaging information video shown or not (in factorial design)) | Sun protection—Sun protection index for behaviours during intentional exposure (self-report), Tanning booth use—Frequency of tanning booth use (self-report); 12 months/Skin darkening; 12 months. Primary outcome data requested and received for sun protection and tanning booth use |
| Mols (2015) [45] | Denmark | Patients referred with chest pain and low to intermediate pretest likelihood of significant coronary artery disease, with an Agatston score of 70 (*n* = 192) | Coronary CT angiography of coronary artery calcification with nurse consultation to discuss the image results and risk factors of coronary artery disease | CT as per intervention but no additional nurse consultation to discuss the image results | Smoking—Current smoking status (self-report), Medication use—Adherent to statins prescribed (self-report), Healthier diet—Index of healthier diet (e.g., daily fruit and vegetable servings) (self-report), Physical activity—Active (>30 minutes moderate intensity activity >3 days of the week) (self-report); 6 months/Systolic blood pressure, diastolic blood pressure, LDL cholesterol, glycated hemoglobin; 6 months |
| Näslund (2019) [46] | Sweden | Individuals from population-based cardiovascular disease prevention programme with 1 or more conventional risk factors, with examinations at hospitals or healthcare centres (*n* = 3,532) | Ultrasound imaging of arterial health provided initially (pictorial representation of carotid ultrasound result plus information for interpretation and advice) and same pictorial information provided again at 6 months | Ultrasound imaging as per intervention but no pictorial representation provided | Smoking—Current smoking status (self-report), Medication use—Use of statins (self-report); 12 months/Framingham risk score, systolic blood pressure, diastolic blood pressure, LDL cholesterol, fasting glucose, waist circumference; 12 months. Diastolic blood pressure was not included in analysis as useable data were not reported |

(*Continued*)

**Table 1.** (Continued)

| Study | Country | Setting and participants (number randomised) | Intervention including type of medical imaging and related health condition | Comparison | Primary outcome(s) selected for review; timing of assessment / secondary outcome(s) selected for review; timing of assessment |
|---|---|---|---|---|---|
| Oliveria (2004) [47] | USA | Outpatient pigmented lesion clinic, with patients with 5 or more clinical dysplastic nevi (*n* = 100) | Whole body photography, used in teaching intervention with photo book featuring photographs and instruction on how to use them to aid skin self-examination of new and changing moles | As intervention but teaching intervention with no photography or photo book, given a written pamphlet on skin self-examination and how to record moles in a diary format | Skin self-examination—Adequate frequency of skin self-examination over past 4 months (self-report); 4 months |
| O'Malley (2003) [48] | USA | Standard periodic cardiovascular screening programme for active duty US Army personnel (*n* = 450) | CT (electron beam tomography) coronary artery screening with results provided (in intensive case management or usual care setting (in factorial design) | CT (electron beam tomography) coronary artery screening with results withheld (in intensive case management or usual care setting (in factorial design) | Smoking—Quitting in smokers (self-report), Physical activity—Index using Baecke Physical Activity questionnaire (self-report); 12 months/ Framingham risk score, systolic blood pressure, diastolic blood pressure, LDL cholesterol, fasting glucose, glycated hemoglobin, BMI, anxiety, depression, stress; 12 months |
| Rees (2013) [49] | Australia | Eye care clinic; diabetic patients with both nonproliferative diabetic retinopathy and suboptimal glycemic control (*n* = 25) | Fundus photography to image retinal health, with orthoptist guiding participants through their own retinal image in contrast with images of a healthy retina and varying degrees of retinopathy, including linking eye health to behaviours | Fundus photography but not shown and guided through images, and completed outcome assessments only | Physical activity—Frequency of physical activity (>30 minute periods) (self-report), Healthier diet—Frequency of following a healthy eating plan (self-report), Foot care—Frequency of checking health of feet in the last 7 days (self-report), Blood glucose testing—Frequency of testing blood sugar in the last 7 days (self-report); 3 months/ Glycated hemoglobin; 3 months. Physical activity and healthier diet outcomes were not included in analyses as final values were not reported and could not be obtained |
| Rodondi (2012) [50] | Switzerland | General population smokers attended research clinic at university (*n* = 536) | Ultrasound imaging of carotid arterial health; if at least 1 carotid atherosclerotic plaque received pictures and 7-minute education on significance, if without atherosclerotic plaques received 7-minute education on smoking risks; smoking cessation programme for 1 year | No ultrasound imaging of carotid arterial health; smoking cessation programme for 1 year | Smoking—Continuous abstinence (biochemically validated), Medication use—Adherence to cardiovascular medication regimen using Morisky medication adherence questionnaire (self-report), Physical activity—Total physical activity using International Physical Activity Questionnaire (self-report); 12 months/Framingham risk score, systolic blood pressure, diastolic blood pressure, LDL cholesterol, depression, stress; 12 months. Primary outcome data requested and received for physical activity |

(*Continued*)

**Table 1.** (Continued)

| Study | Country | Setting and participants (number randomised) | Intervention including type of medical imaging and related health condition | Comparison | Primary outcome(s) selected for review; timing of assessment / secondary outcome(s) selected for review; timing of assessment |
|---|---|---|---|---|---|
| Rozanski (2011) [51] | USA | Medical centre middle-aged individuals with coronary artery disease risk factors (n = 2,137) | CAC scanning using CT, reviewed images and score with nurse and received copies of scan report, plus reviewed guidelines on cardiac risk factors with nurse | No CAC scanning, reviewed guidelines on cardiac risk factors with nurse | Smoking—Quitting in smokers (self-report), Medication use—Adherence to lipid-lowering meds (self-report), Physical activity—Exercise >3 times/week in nonexercisers (self-report); 48 months/Framingham risk score, systolic blood pressure, diastolic blood pressure, LDL cholesterol, fasting glucose, waist circumference; 48 months |
| Shahab (2007) [52] | UK | Smokers attending cardiovascular outpatient clinic (n = 23) | Ultrasound imaging of carotid arterial health plus verbal feedback from a cardiovascular consultant with photographs contrasting a healthy artery and their own arteries | Ultrasound imaging of carotid arterial health but with verbal feedback only | Smoking—Incidence of smoking cessation behaviours (self-report); 1 month |
| Stock (2009) [58] | USA | Male outdoor workers at organisational offices (n = 148) | UV photography for sun damage, with skin damage explained in relation to comparison between natural light and UV photos (and given educational video or not (in factorial design)) | As intervention but no UV photography, with only natural light photo shown (and given educational video or not (in factorial design)) | Sun protection—Sun protection behaviours index (self-report and objective); 12 months |
| Whitehead (2014) [59] | UK | Laboratory study with university students/staff (n = 73) | Photography to image skin health including leaflet containing photos, manipulated with spectrophotometry-derived effect to show effect of fruit and vegetable consumption, plus information about fruit and vegetable consumption and health | No photography, with information about fruit and vegetable consumption and health only | Healthier diet—Fruit and vegetable consumption (self-report); 10 weeks. Primary outcome data requested and received for healthier diet |

BMI, body mass index; CAC, coronary artery calcium; CT, computed tomography; HS&DR, Health Services & Delivery Research; LDL, low-density lipoprotein; NIHR, National Institute for Health Research; UV, ultraviolet.

**Healthier diet.** Five studies [40,43–45,59] assessed self-reported dietary behaviours, with follow-up of up to 12 months, this being the mode. Three studies evaluated either ultrasound or CT to visualise cardiovascular health [40,44,45], one used photography to image skin health related to fruit and vegetable consumption [59], and one study used quantitative light fluorescence photography to visualise oral health [43]. Pooled analysis (n = 467) showed that imaging feedback led to healthier dietary behaviour (SMD 0.30, 95% CI 0.11 to 0.50, p = 0.003, $I^2$ = 0% ($I^2$ CI 0% to 64%); Fig 3). Evidence for this outcome was assessed to be of low certainty using the GRADE approach.

**Medication use.** Six studies [31,35–37,41,42] assessed use of cardiovascular medications, predominantly statins, with follow-up of up to 48 months, 12 months being the mode. All six studies used either ultrasound or CT to visualise cardiovascular health. Pooled analysis (n = 4,539) showed that imaging feedback had no impact upon medication use (SMD 0.04, 95% CI −0.24 to 0.32, p = 0.79, $I^2$ = 85% ($I^2$ CI 64% to 91%); Fig 3). Evidence for this outcome was assessed to be of very low certainty using the GRADE approach.

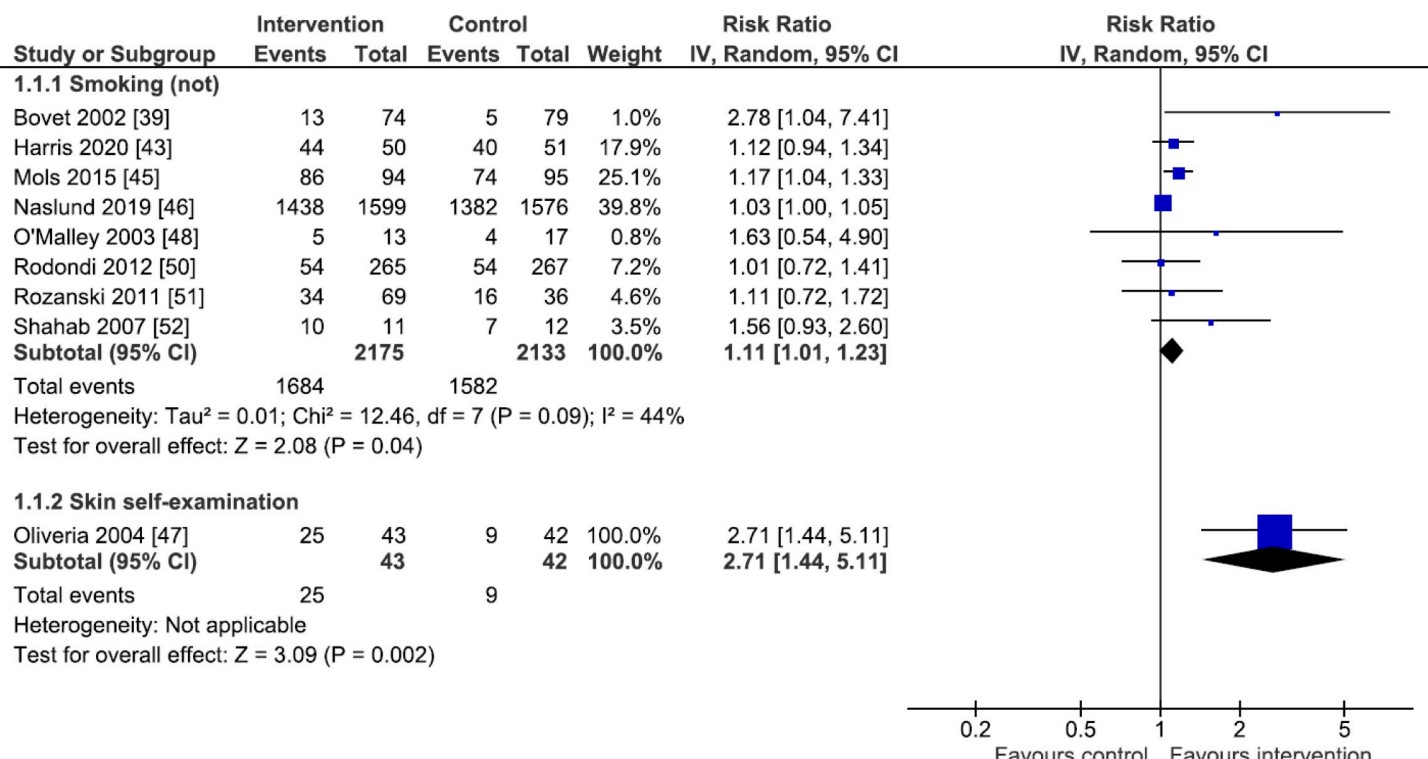

**Fig 2. Primary outcome analysis: Smoking and skin self-examination.** Forest plots are presented for meta-analyses summing the effects of contributing studies for each outcome. In each forest plot, effect estimates from individual studies are illustrated with a box and the 95% CIs with lines (whiskers). The overall effect is indicated by the diamond below, with its width representing the 95% CIs (any overlap of the central line of no effect indicates no statistically significant difference between the intervention and control groups). CI, confidence interval; IV, inverse variance.

In exploring the substantial heterogeneity observed for this outcome, characteristics of the constituent studies were heterogenous for outcome time point, outcome measure, and the nature of the control group. While differences in the former two characteristics did not significantly explain heterogeneity, for the latter, removing three studies [40,45,46] that differed in their control group—with control participants undergoing (versus not undergoing) the medical imaging procedure—markedly reduced values for $I^2$ (from 85% to 0%) and chi-squared (from 32.35 ($p < 0.001$) to 1.66 ($p = 0.44$)). This analysis suggests that the nature of the control group may be contributing to the heterogeneity observed in the main meta-analysis. None of these sensitivity analyses meaningfully altered the meta-analysis result or its interpretation.

**Sun protection.** Five studies [54–58] assessed self-reported sun protection behaviours up to 12 months postintervention, with 12 months being the modal time of assessment. All five studies used UV photography to image UV exposure–related skin damage. Pooled analysis ($n = 487$) showed no statistically significant effect of imaging feedback on sun protection behaviours (SMD 0.14, 95% CI −0.04 to 0.32, $p = 0.12$, $I^2 = 0\%$ ($I^2$ CI 0% to 64%); Fig 4). Evidence for this outcome was assessed to be of low certainty using the GRADE approach.

**Oral hygiene behaviours.** Three studies assessed self-reported oral hygiene behaviours (toothbrushing and flossing) at four [41], eight [42], or 12 months [43]. The former two studies used intraoral camera photography to visualise oral health, while the latter study used quantitative light fluorescence photography. Pooled analysis ($n = 321$) showed that imaging feedback increased oral hygiene behaviours (SMD 0.35, 95% CI 0.13 to 0.57, $p = 0.002$, $I^2 = 0\%$ ($I^2$ CI 0%

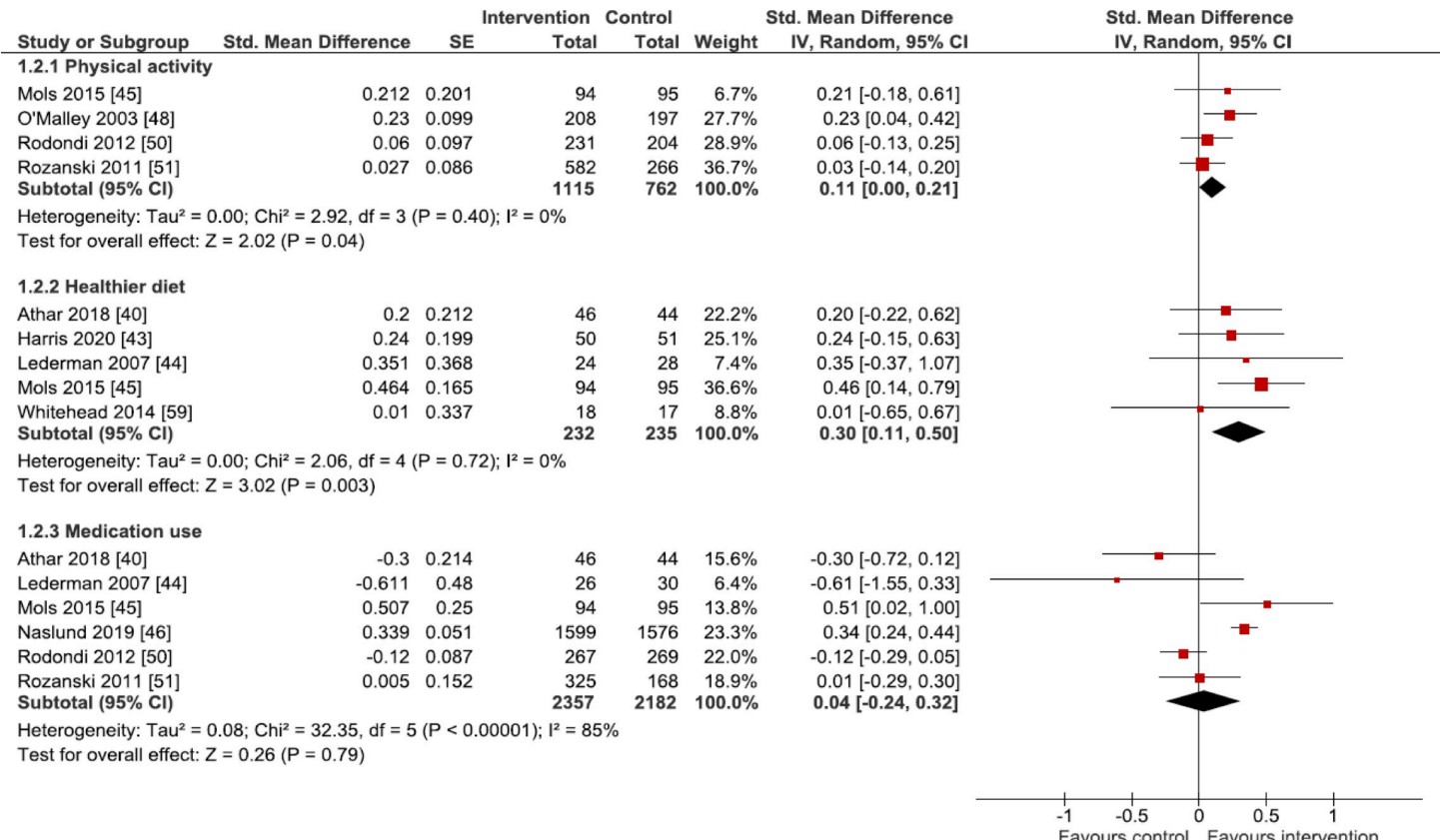

**Fig 3. Primary outcome analysis: Physical activity, diet, and medication use.** Forest plots are presented for meta-analyses summing the effects of contributing studies for each outcome. In each forest plot, effect estimates from individual studies are illustrated with a box and the 95% CIs with lines (whiskers). The overall effect is indicated by the diamond below, with its width representing the 95% CIs (any overlap of the central line of no effect indicates no statistically significant difference between the intervention and control groups). CI, confidence interval; IV, inverse variance; SE, standard error.

to 73%); Fig 4). Evidence for this outcome was assessed to be of low certainty using the GRADE approach.

**Tanning booth use.**   Two studies assessed self-reported frequency of tanning booth use at one month [53] and at 12 months [57]. Both studies used UV photography to image UV exposure–related skin damage. Pooled analysis ($n$ = 465) showed no clear effect of imaging feedback (SMD 0.27, 95% CI −0.15 to 0.68, $p$ = 0.21, $I^2$ = 73%; Fig 4). Evidence for this outcome was assessed to be of very low certainty using the GRADE approach. In exploring the substantial heterogeneity observed for this outcome, characteristics of the two constituent studies were heterogenous only in outcome time point, meaning this characteristic may be contributing to the observed heterogeneity.

**Foot care.**   A single small study [49] (analysed $n$ = 25) showed that feedback of fundus photography to image retinal health linked to diabetes increased reported foot care behaviour at three months (SMD 1.03, 95% CI 0.17 to 1.89, $p$ = 0.02; Fig 4). Evidence for this outcome was assessed to be of low certainty using the GRADE approach.

**Blood glucose testing.**   A single small study [49] (analysed $n$ = 25) showed no clear evidence that feedback of fundus photography to image retinal health linked to diabetes impacted upon blood glucose testing behaviour at three months (SMD 0.30, 95% CI −0.50 to 1.11, $p$ = 0.46; Fig 4). Evidence for this outcome was assessed to be of very low certainty using the GRADE approach.

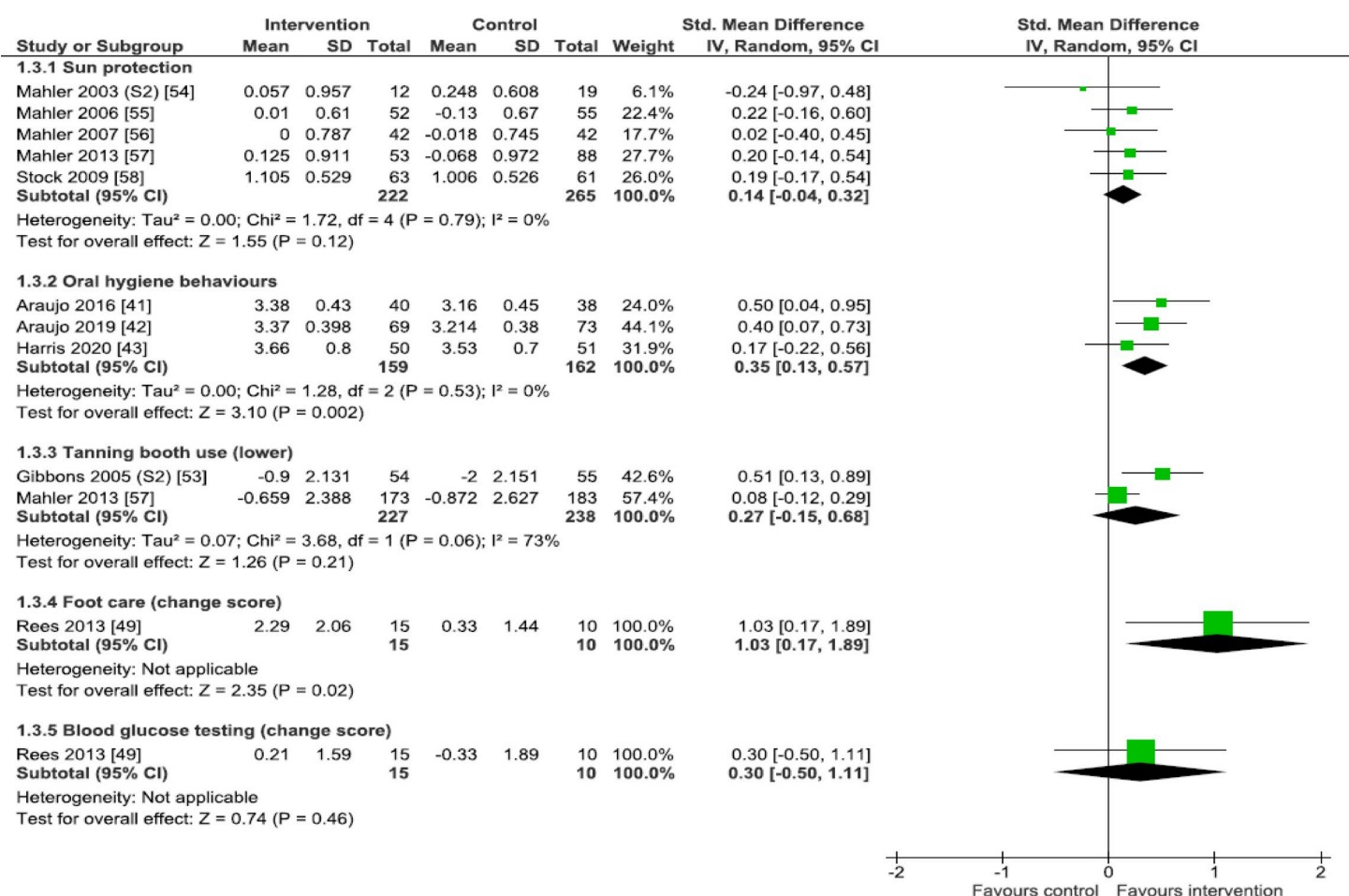

**Fig 4. Primary outcome analysis: Sun protection, oral hygiene behaviours, tanning booth use, foot care, and blood glucose testing.** Forest plots are presented for meta-analyses summing the effects of contributing studies for each outcome. In each forest plot, effect estimates from individual studies are illustrated with a box and the 95% CIs with lines (whiskers). The overall effect is indicated by the diamond below, with its width representing the 95% CIs (any overlap of the central line of no effect indicates no statistically significant difference between the intervention and control groups). CI, confidence interval; IV, inverse variance; SD, standard deviation.

## Secondary outcomes

Separate forest plots display the results for risk factors and mental health outcomes where the effect sizes are mean differences (Fig 5) and where they are SMDs (Fig 6). Visual feedback was associated with reductions in systolic blood pressure with a modal time point of 12 months (mean difference −1.58 mm Hg, 95% CI −2.41 to -0.74, $p < 0.001$, $n = 6123$, $I^2 = 6\%$ ($I^2$ CI 0% to 63%); Fig 5) and reductions in waist circumference at 12 or 48 months (mean difference −1.87 cm, 95% CI −3.19 to −0.56, $p = 0.005$, $n = 5015$, $I^2 = 75\%$; Fig 5), as well as with improvements in oral health at four, eight, or 12 months (SMD −0.54, 95% CI −0.88 to −0.19, $p = 0.002$, $n = 321$, $I^2 = 56\%$ ($I^2$ CI 0% to 86%); Fig 6). There was some indication that feedback reduced Framingham risk score with a modal time point of 12 months, but with uncertainty due to the 95% CI slightly overlapping no effect (mean difference −0.44, 95% CI −0.95 to 0.06, $p = 0.08$, $n = 5923$, $I^2 = 70\%$ ($I^2$ CI 0% to 88%); Fig 5). There was no clear evidence of beneficial or detrimental effects on any other secondary outcomes, namely diastolic blood pressure (modal time point of 12 months), LDL cholesterol (modal time point of 12 months), fasting glucose (modal time point of 12 months), glycated hemoglobin (modal time point of 12 months), BMI at 12 months, or skin darkening due to UV exposure at two or 12 months. Although rarely assessed,

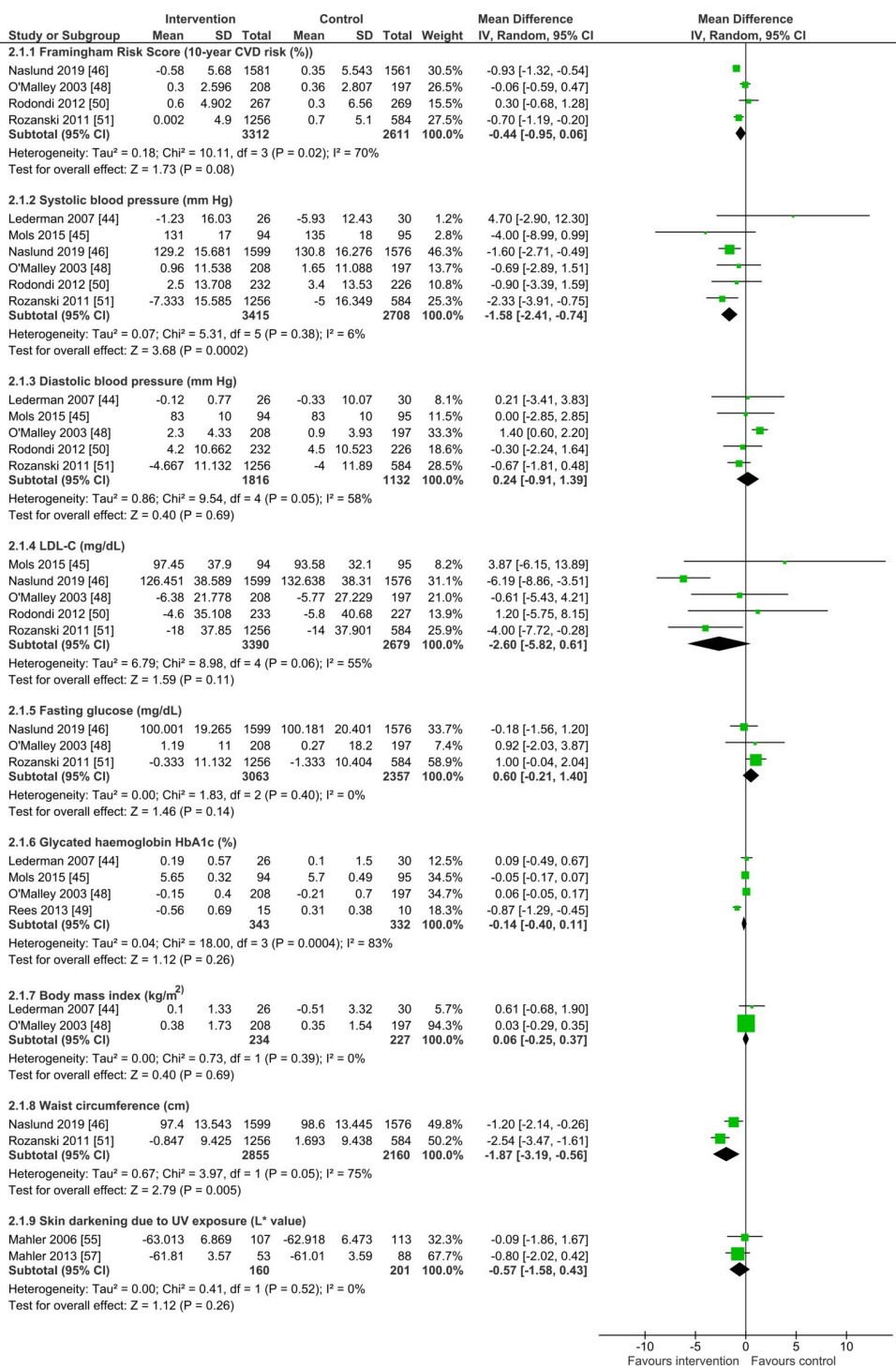

**Fig 5. Secondary outcome analysis (outcomes with standard scales): Framingham risk score, systolic blood pressure, diastolic blood pressure, LDL cholesterol, fasting glucose, glycated hemoglobin, BMI, waist circumference, and skin darkening.** Forest plots are presented for meta-analyses summing the effects of contributing studies for each outcome. In each forest plot, effect estimates from individual studies are illustrated with a box and the 95% CIs with lines (whiskers). The overall effect is indicated by the diamond below, with its width representing the 95% CIs (any overlap of the central line of no effect indicates no statistically significant difference between the intervention and control groups). BMI, body mass index; CI, confidence interval; CVD, cardiovascular disease; IV, inverse variance; LDL, low-density lipoprotein; SD, standard deviation; UV, ultraviolet.

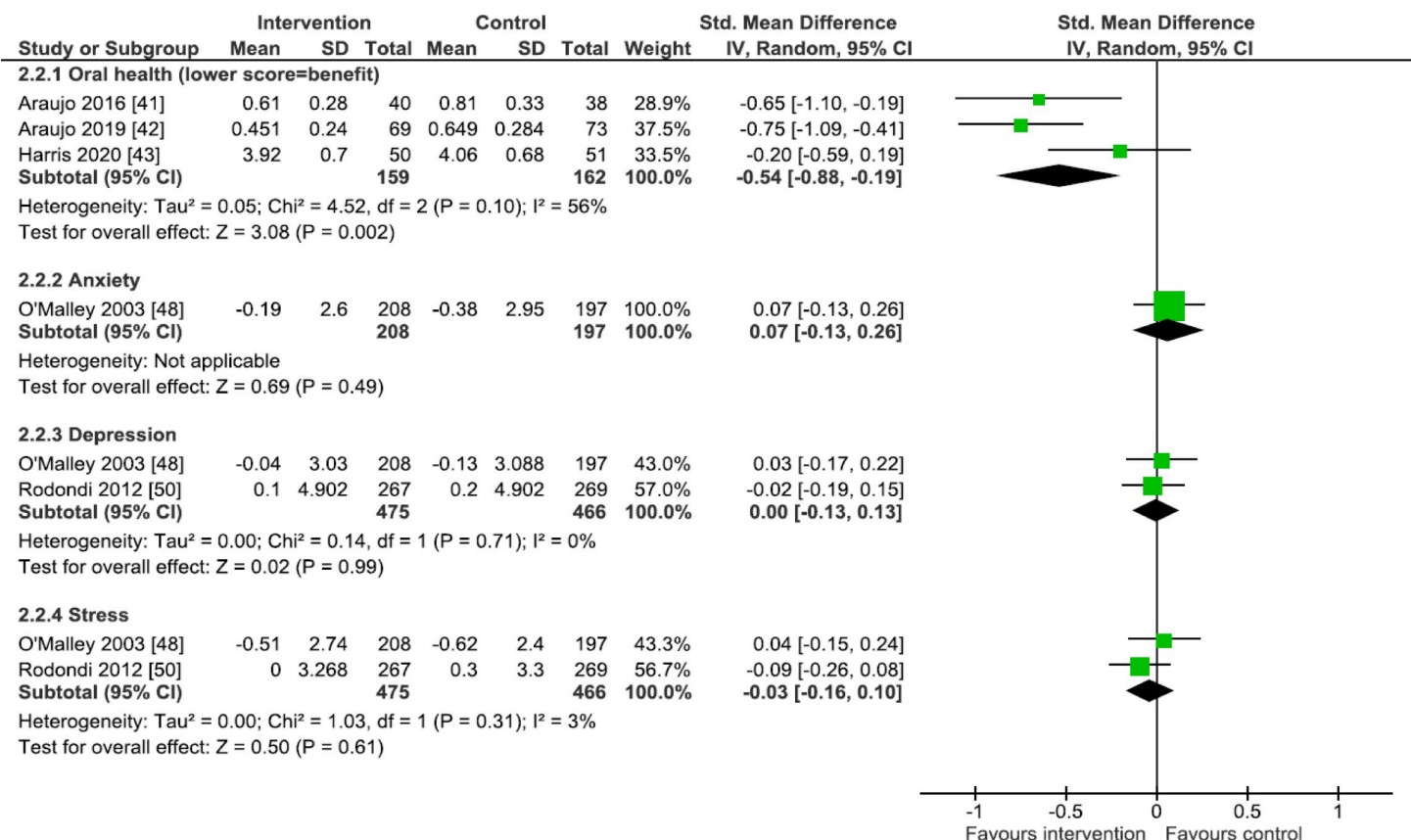

**Fig 6. Secondary outcome analysis (outcomes with nonstandard scales): Oral health, anxiety, depression, and stress.** Forest plots are presented for meta-analyses summing the effects of contributing studies for each outcome. In each forest plot, effect estimates from individual studies are illustrated with a box and the 95% CIs with lines (whiskers). The overall effect is indicated by the diamond below, with its width representing the 95% CIs (any overlap of the central line of no effect indicates no statistically significant difference between the intervention and control groups). CI, confidence interval; IV, inverse variance; SD, standard deviation.

there was no evidence of any effects on mental health, namely anxiety, depression, or stress, all assessed at 12 months (Fig 6), and no other reported harms linked to the interventions.

The substantial heterogeneity observed for some of these secondary outcomes—namely Framingham risk score, diastolic blood pressure, LDL cholesterol, glycated hemoglobin, waist circumference, and oral health—was explored. Regarding Framingham risk score (Fig 5), the characteristics of the constituent studies were heterogenous regarding outcome time point and the nature of the control group. These differences did not significantly explain heterogeneity, nor did they meaningfully impact upon the meta-analysis result or its interpretation. For diastolic blood pressure (Fig 5), the characteristics of the constituent studies were heterogenous for outcome time point and the nature of the control group. For outcome time point, removing two studies [45,51] that did not assess the outcome at 12 months, reduced heterogeneity as determined by values for $I^2$ (from 58% to 28%) and chi-squared (from 9.54($p = 0.05$) to 2.77 ($p = 0.25$)). For the nature of the control group, removing two studies [45,48] that differed in this regard (with control participants undergoing the medical imaging procedure) markedly reduced heterogeneity, with $I^2$ decreasing from 58% to 0% and chi-squared decreasing from 9.54 ($p = 0.05$) to 0.27 ($p = 0.87$). This analysis suggests that both of these factors may be contributing to the heterogeneity observed in the main meta-analysis. Neither of these analyses meaningfully altered the meta-analysis result nor its interpretation. Regarding LDL cholesterol (Fig 5), the characteristics of the constituent studies were heterogenous regarding outcome

time point and the nature of the control group. These differences did not significantly explain heterogeneity, nor did they meaningfully impact upon the meta-analysis result or its interpretation.

Regarding glycated hemoglobin (Fig 5), the characteristics of the constituent studies were heterogenous for outcome time point, the health condition being imaged, and the nature of the control group. For outcome time point, removing two studies [45,49] that did not assess the outcome at 12 months markedly reduced heterogeneity as determined by values for $I^2$ (from 83% to 0%) and chi-squared (from 18.00 ($p < 0.001$) to 0.01 ($p = 0.92$)). For the health condition being imaged, excluding one study [49] that conducted retinal (versus cardiovascular) imaging markedly reduced heterogeneity with $I^2$ reducing from 83% to 0% and chi-squared from 18.00 ($p < 0.001$) to 1.84 ($p = 0.40$). The nature of the control group did not explain heterogeneity. This analysis suggests that both outcome time point and the health condition being imaged may be contributing to the heterogeneity observed in the main meta-analysis. None of these analyses meaningfully altered the meta-analysis result or its interpretation. Regarding waist circumference (Fig 5), the characteristics of the two studies were heterogenous regarding both outcome time point and the nature of the control group, meaning these characteristics may be contributing to the observed heterogeneity. Finally, for oral health (Fig 6), the characteristics of the constituent studies were heterogenous for outcome time point, and outcome measure. For the outcome measure, removing the study that measured toothbrushing only [43], leaving two studies [41,42] that measured a composite of toothbrushing and flossing, markedly reduced heterogeneity as determined by values for $I^2$ (from 56% to 0%) and chi-squared (from 4.52 ($p = 0.10$) to 0.12 ($p = 0.73$)). This analysis suggests that the outcome measure may be contributing to the heterogeneity observed in the main meta-analysis. It did not meaningfully alter the meta-analysis result or its interpretation. The impact of heterogeneity of outcome time point could not be explored because there was no modal time point for the constituent studies.

## Assessment of risk of bias and certainty of evidence

None of the 21 studies were judged to have a low summary risk of bias for the primary outcomes (Fig 7). This reflected both a lack of clarity in reporting and an inability or failure to safeguard against risk of bias. The judgement that there were some concerns about risk of bias was for most studies determined by them including only or predominantly self-report measures of risk-reducing behaviours—applying to all but one study [50]—and not making prespecified analysis plans available, with only two studies appearing to do this [43,46]. In terms of GRADE assessment of the certainty of the evidence for each of the primary outcomes, smoking was the only outcome judged to have evidence of moderate certainty (downgraded once only for study limitations due to risk of bias). For other primary outcomes, evidence was determined to be of low (skin self-examination, physical activity, healthier diet, sun protection, oral hygiene behaviours, foot care; and blood glucose testing) or very low (medication use and tanning booth use) certainty. All of these outcomes were downgraded once for study limitations due to all contributing studies being at risk of bias. Other than for smoking (which had a large sample size of over 4,000 participants and a clear effect not overlapping zero), evidence for all outcomes was downgraded again due to imprecision, with the sample size not meeting the optimal information size and/or 95% CIs for the summary effect estimate encompassing both harm and benefit. Other than that of smoking, the only analysis that included more than 2,000 participants was for medication use (but its effect estimate encompassed both notable harm and benefit), and other than physical activity, all other primary outcome analyses included fewer than 500 participants. For medication use and tanning booth use only, evidence

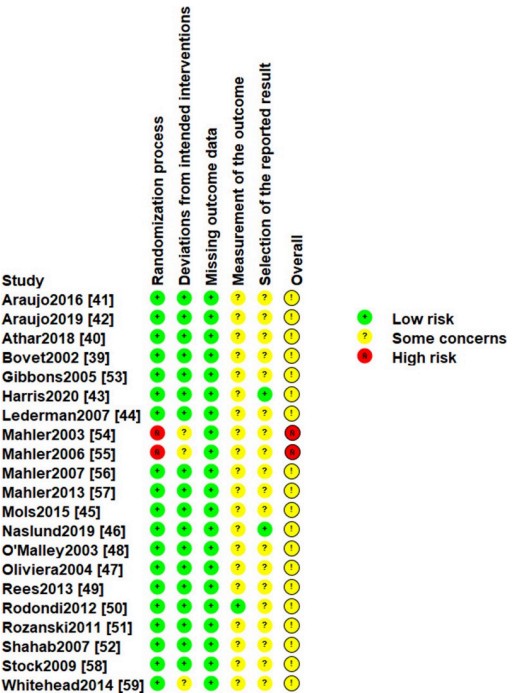

**Fig 7. Risk of bias assessment.** Judgements concerning risk of bias for primary outcomes are presented for each study in accordance with the RoB 2 tool, including an "Overall" summary judgement (Low risk, Some concerns, and High risk).

was additionally downgraded for inconsistency, with substantial heterogeneity identified, as determined by a value of 50% or greater for the $I^2$ statistic (this value being 85% for medication use and 73% for tanning booth use).

## Assessing clinical relevance of effect estimates

For specific primary outcomes—smoking, healthier diet, and oral hygiene behaviours—for which a meta-analysis of $\geq$3 studies indicated an at least small (e.g., SMD of $\geq$0.2) and statistically significant effect, we additionally sought to interpret the observed effect sizes in terms of the potential for a clinically significant or meaningful impact. In addition, we similarly interpreted the meta-analytic effect size estimates for one clinically important secondary outcome, namely systolic blood pressure. Where possible these interpretations were made by reference to published guidance that specifically concerned determining clinical significance within that domain, or to relevant large-scale data for that outcome outwith the data included in the review. If such information was not available, we extrapolated from related data from the largest and lowest risk of bias trial within the given meta-analysis in the current review. These selected extrapolations are intended to be illustrative only, being neither comprehensive nor definitive. This reflects that the review focuses principally on considering the set of findings holistically to inform broad approaches rather than seeking to generate guidance for any specific intervention context. They require considerable caution as they inevitably involve assumptions about the summarised effect in and of itself, as well as about how clinical significance can be understood in light of those data.

Regarding the result of the meta-analysis for smoking (RR of 1.11), in clinical trials of smoking cessation treatments, intervention effects of above 7% to 9% (but also frequently those smaller than that) for abstinence of at least 6 months are widely considered clinically

significant, such as by licensing authorities evaluating therapies [60]. This suggests that the intervention has the potential to have clinically meaningful effects and is in light of evidence that for those who quit smoking, within 5 years, their risk of death from all causes reduces by 13% relative to those who continue to smoke [61]. For healthier eating, given the result of the meta-analysis (SMD = 0.30), if an effect of comparable magnitude was seen for calories consumed, with adults' mean (±SD) daily energy intake from food estimated as 1,773(± 561) kcal [62], then this suggests a potentially meaningful reduction in daily calories of 168 kcal or approximately 9% (168/1,773 kcal). This is in light of estimates that a daily reduction of approximately 28 kcals per person would prevent further weight gain in 90% of the population in England [63,64]. Regarding the results of the meta-analysis for oral hygiene behaviours (SMD = 0.35), based on baseline data from a study included in this meta-analysis [43] for teeth brushing duration in minutes: mean(SD) = 3.20(0.69), the magnitude of the intervention effect could be extrapolated to an increase in brushing duration of approximately 14 seconds, an approximately 7% increase. This appears a valuable reduction given evidence of a clear relationship between brushing duration and plaque removal [65]. Finally, for systolic blood pressure, given evidence that every 10 mm Hg increment of systolic blood pressure is associated with a 5% increased risk of cardiovascular events [66], the meta-analytic effect from this review of −1.58 mm Hg suggests that the intervention could reduce risk of cardiovascular events by approximately 0.79% (Number Needed to Treat = 127). While this effect appears small, it could be considered notable given the potential for these interventions to both take advantage of existing procedures and data and be relatively scalable (see Discussion).

## Discussion

### Principal findings

The results of this review highlight the potential for visualisation and feedback of health risk using medical imaging to motivate risk-reducing changes in a range of health-related behaviours. There was evidence of benefit for six of 10 behavioural outcomes analysed, and no evidence of adverse effects, with the direction of effect for all behaviours favouring the intervention to some degree. Of the primary behavioural outcomes, however, only evidence for reduced smoking was judged to be of at least moderate certainty. Evidence for all other behaviours was of low or occasionally very low certainty—meaning we can be less certain that the true effect will not substantially differ—although the overall pattern of results across outcomes partly allays this concern. The broadly consistent findings for risk factor outcomes, with several beneficial effects observed and no evidence for adverse impacts, provides some further support for the behavioural findings. For risk factors, evidence was strongest for a reduction in systolic blood pressure, featuring a large sample (>6,000 participants), minimal heterogeneity, and—unlike for behavioural outcomes that typically rely on self-report measures—standardised objective measurement.

The analyses typically covered extended follow-up periods, with the modal time point for outcome assessment being 12 months for outcomes including smoking, physical activity, healthier diet, medication use, and sun protection behaviours. This highlights the potential for effects elicited by these interventions to be sustained in the medium to long term. Beyond the data from studies included in this review, there is complementary evidence that risk-reducing behavioural changes such as changes to smoking, physical activity, or diet—whether occurring within intervention contexts or when examined longitudinally in free-living conditions—can be well maintained over time [67]. For example, it has been estimated that for smokers who remain abstinent at 12 months, the annual incidence of relapse after this is only around 10% [68].

This review adds to existing research through a comprehensive, systematic approach to consolidate the experimental evidence for the potential impact of providing visual feedback of medical imaging. Its findings are broadly consistent with those from previous systematic reviews with an overlapping intervention focus. For example, earlier reviews have highlighted the potential of highly salient health risk feedback that can indicate existing bodily damage and active disease progression, such as assessments of coronary arterial calcification, to motivate changes in recipient behaviour [28,29]. By contrast, previous reviews that focus on the effects of providing more abstract genetic or phenotypic personalised risk information that often concerns risks of future events suggest that this has a weak or nonexistent impact on recipients' health-related behaviours [3,4,6,7]. We consider it plausible—as outlined in the Introduction—that imaging interventions could be more potent, at least in more at-risk populations likely to receive more concerning feedback. Study populations within the review were typically at greater baseline risk than the general population, with known clinical or behavioural risk factors such as having a serious health condition, meeting risk marker thresholds, or being a smoker. While only a small number of studies [46,48,51] formally assessed whether the degree of disease that was communicated moderated the intervention effect, they did report greater benefits on cardiovascular risk outcomes for participants who received feedback illustrating more (versus less) advanced arterial disease. The broad set of contexts included also highlight the possibility of a relatively generalisable principle that could be applicable across a range of treatment contexts.

## Strengths and limitations of this review

We conducted the review using rigorous Cochrane methods [38] to minimise the risk of bias. We identified and quantitatively synthesised a substantive body of participant data from randomised controlled trials with typically lengthy follow-up periods and strengthened our approach with systematic assessment of risk of bias of included studies [34] and of certainty of the evidence by outcome [33]. Previous reviews had identified relatively few clinical studies using randomised designs [32] or were narrowly focused on single behaviours or treatment domains.

Our review has several important limitations, some linked to limitations of the available evidence. First, nearly all included studies were judged to have at least some concerns for risk of bias for primary outcomes. In particular, the widespread failure or inability to use valid precise measures of behaviour may have introduced error and bias. While we acknowledge the logistical challenges and potential for measurement reactivity associated with objective measures of behaviour [69] and that the use of self-report measures is sometimes necessary, included studies often used self-report measures even when viable objective measures were available (for example, in relation to smoking cessation and physical activity). As participants and providers are not blinded to these interventions, it is important that outcome assessors are, but self-report measures preclude this and also add imprecision. We note, however, that this concern largely does not apply to the complementary evidence of impact on risk factors. In addition, the potential for selective outcome reporting was notable, with very few instances of preregistered protocols and detailed analysis plans. Concerns about risk of bias contribute to the low certainty assessed for most results, in conjunction with other issues including the presence of several small, likely underpowered studies that limit the power and precision of meta-analyses. We were also unable to formally examine the likelihood of publication bias due to the small numbers of studies for each meta-analysis, although we sought to minimise its possible impact by searching for grey literature.

Second, we were unable to systematically interrogate observed heterogeneity to determine potentially important factors that may modify effects, such as via subgroup or meta-regression analyses, due to insufficient data for any single review outcome. Future versions of this review

with more power should plan to conduct such analyses, in particular for disentangling the potential for specific characteristics or components of intervention and control procedures to modify the observed effects. Third, for some meta-analyses, multiple studies were conducted by the same or considerably overlapping groups of researchers—this was marked for sun protection behaviours, tanning booth use, and oral hygiene behaviours (applying to 5 of 5, 2 of 2, and 2 of 3 studies, respectively). In these cases, the individual studies cannot be assumed to be wholly independent, lowering confidence in the pooled analyses that were already judged to be of low (in the case of sun protection and oral hygiene behaviours) or very low (tanning booth use) certainty. Finally, this review largely concerns direct effects of visualised feedback on its recipients - typically patients - and therefore does not assess effects on healthcare systems and practitioners' behaviours. These can have direct or indirect effects on patient outcomes, as patient behaviours may be influenced by discussion with practitioners, and some—such as medication use—will require and reflect actions taken by practitioners.

## Implications for healthcare practice and research

The magnitude of the beneficial effects on primary and secondary outcomes appears consistent with the relatively small effects observed for other comparable nonpharmaceutical interventions that are widely implemented and regarded as having potentially clinically important impacts. These include effects of behavioural support, cardiovascular risk assessment, and health checks, on reducing smoking, increasing physical activity, and improving risk factor outcomes [70–73]. Effects of small magnitudes could therefore be reasonably expected, in particular because the interventions were typically relatively brief and delivered at single time points. However, even modest effects are potentially important should they be scalable to the large and growing number of people being investigated with medical imaging.

While for the more notable results we have attempted to interpret the findings in relation to their possible clinical significance, including some reference to morbidity end points, the purpose of such extrapolations cannot be to provide definitive evidence of the potential effect on health outcomes. It is instead to provide some additional context for understanding the results as well as to illustrate how one might attempt to interpret these and other effect sizes from this review. Such extrapolations are necessarily highly tentative and indirect as the included evidence does not in itself address the entire causal sequence from intervention through to behaviour and through to harder clinical end points such as morbidity or premature mortality. In general, it is not expected that studies in this area that have a substantive focus on behavioural impacts would also include distal health outcome data, although more proximal risk factors are often measured. It may be that any inferences drawn about harder clinical end points can only ever be in the general sense that even small changes in behaviours and risk factors could have important implications given robust associations with morbidity and mortality outcomes [2].

Any assessments of the potential clinical significance of the observed effects will also need to be made in relation to associated costs and disbenefits of the intervention, which appear to be relatively small. While medical imaging is in itself costly, it is usually only pursued in those who require it due to a known or probable health risk, or as part of a routine screening programme. Adding visualised feedback as standard for individuals who are already being imaged may represent a relatively simple procedural change that capitalises efficiently on this existing widespread opportunity, and could outperform more typical health risk communications and advice. Although it potentially requires some additional practitioner time, it is also feasible that it could be delivered in a predominantly automated, standardised manner. However, stronger evidence of effectiveness would likely be needed to justify new medical imaging procedures being undertaken with the primarily therapeutic purpose of motivating behavioural

changes in general populations without known risk factors and not already engaged with clinical services. It is noted, however, that as these technologies become cheaper and more widespread, screening of larger populations at lesser risk may become more justifiable as a form of primary prevention, albeit still being dependent on evidence of benefit.

Future research priorities to build this evidence base and corroborate current findings centre on the need for additional adequately-powered trials that better mitigate against risk of bias, in particular through using objective measures of behaviour and risk factors, and registering detailed protocols and prespecified statistical analysis plans. Although in recent years some large-scale randomised controlled trials have been conducted in this area, evidence remains limited for some key behavioural outcomes: of the primary outcome analyses, only medication use, smoking, and physical activity involved notably large samples. It is encouraging in this regard that multiple ongoing studies were identified within the review process which appear likely to be able to contribute to a future update of this review [74–76]. In addition, detailed work is needed to elucidate the psychological mechanisms that underlie or modify observed effects on behaviour, both in general and within specific treatment contexts [22], as well as to disentangle the contributions of many visual and verbal components, techniques, and interactions that form the inherently complex intervention packages that are delivered. A deepened mechanistic understanding of both the intervention's active characteristics and its effects would have the potential to optimise its design and application.

## Conclusions

In this study, we observed that feedback of medical images to individuals has the potential to motivate risk-reducing behaviours and reduce risk factors. Should this promise be corroborated through further adequately powered trials that better mitigate against risk of bias, such interventions could usefully capitalise upon widespread and growing use of medical imaging technologies in healthcare.

## Supporting information

**S1 Text.** Review protocol (Appendix A); Database search strategies (Appendix B); Outcome selection process (Appendix C).
(DOCX)

**S1 Checklist. PRISMA checklist.** PRISMA, Preferred Reporting Items for Systematic Reviews and Meta-Analyses.
(DOCX)

## Acknowledgments

We are grateful to the authors of included studies who responded to our requests for further data and to Stephen Sharp for guidance on calculating $I^2$ confidence intervals.

## Author Contributions

**Conceptualization:** Gareth J. Hollands.

**Data curation:** Gareth J. Hollands, Rana Hasan, Florence Alexander.

**Formal analysis:** Gareth J. Hollands.

**Investigation:** Gareth J. Hollands, Juliet A. Usher-Smith, Rana Hasan, Florence Alexander, Simon J. Griffin.

**Methodology:** Gareth J. Hollands, Juliet A. Usher-Smith, Simon J. Griffin.

**Project administration:** Gareth J. Hollands, Juliet A. Usher-Smith, Rana Hasan, Florence Alexander, Natasha Clarke, Simon J. Griffin.

**Supervision:** Gareth J. Hollands, Juliet A. Usher-Smith, Simon J. Griffin.

**Validation:** Natasha Clarke.

**Writing – original draft:** Gareth J. Hollands.

**Writing – review & editing:** Gareth J. Hollands, Juliet A. Usher-Smith, Rana Hasan, Florence Alexander, Natasha Clarke, Simon J. Griffin.

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
