## [Editor Report · Decision Letter 0]

9 Mar 2021

Dear Dr Hollands, 

Thank you for submitting your manuscript entitled "Impact of visualising health risk with medical imaging on recipients’ health-related behaviours and risk factors: systematic review with meta-analysis" for consideration by PLOS Medicine.

Your manuscript has now been evaluated by the PLOS Medicine editorial staff as well as by an academic editor with relevant expertise and I am writing to let you know that we would like to send your submission out for external peer review.

Please be sure that any associated appendix or supporting information file is also included.

Kind regards,

Caitlin Moyer, Ph.D.

Associate Editor

PLOS Medicine

---

## [Decision Letter · Decision Letter 1]

23 Jul 2021

Dear Dr. Hollands,

Thank you very much for submitting your manuscript "Impact of visualising health risk with medical imaging on recipients’ health-related behaviours and risk factors: systematic review with meta-analysis" (PMEDICINE-D-21-01143R1) for consideration at PLOS Medicine. 

Your paper was evaluated by a senior editor and discussed among all the editors here. It was also discussed with an academic editor with relevant expertise, and sent to four independent reviewers, including a statistical reviewer. The reviews are appended at the bottom of this email and any accompanying reviewer attachments can be seen via the link below:

[LINK]

In light of these reviews, I am afraid that we will not be able to accept the manuscript for publication in the journal in its current form, but we would like to consider a revised version that addresses the reviewers' and editors' comments. Obviously we cannot make any decision about publication until we have seen the revised manuscript and your response, and we plan to seek re-review by one or more of the reviewers. 

We expect to receive your revised manuscript by Aug 13 2021 11:59PM. Please email us (plosmedicine@plos.org) if you have any questions or concerns.

We look forward to receiving your revised manuscript. 

Sincerely,

Caitlin Moyer, Ph.D.

Associate Editor 

PLOS Medicine

plosmedicine.org

1. Throughout: Please include line numbers with the revised version.

2. Abstract: Please structure your abstract using the three PLOS Medicine headings (Background, Methods and Findings, Conclusions).

3. Abstract: Please include both 95% CIs and p values for each outcome described in the abstract for the meta-analysis (the p value is misisng for physical activity).

4. Abstract: In the last sentence of the Abstract Methods and Findings section, please describe the main limitation(s) of the study's methodology.

5. Abstract: Conclusions: Please address the study implications without overreaching what can be concluded from the data; the phrase "In this study, we observed ..." may be useful.

6. Author Summary: At this stage, we ask that you include a short, non-technical Author Summary of your research to make findings accessible to a wide audience that includes both scientists and non-scientists. The Author Summary should immediately follow the Abstract in your revised manuscript. This text is subject to editorial change and should be distinct from the scientific abstract. Please see our author guidelines for more information: https://journals.plos.org/plosmedicine/s/revising-your-manuscript#loc-author-summary

7. Methods: Please include all details necessary to understand the methods and analyses, although you note they have been published elsewhere.

8.Methods: Protocol: Please include the prospective protocol as a supporting information file.

9. Methods: Please update the search to the present time (currently December 2019).

10. Methods: Please report your SR/MA according to the PRISMA guidelines provided at the EQUATOR site.

http://www.equator-network.org/reporting-guidelines/prisma/

Please provide the completed PRISMA checklist.

Please add the following statement, or similar, to the Methods: "This study is reported as per the Preferred Reporting Items for Systematic Reviews and Meta-Analyses (PRISMA) guideline (S1 Checklist)."

11. Results: Please report p values to two decimal places, and three decimal places for p values between .01 and .001. Please use p<.001 for values less than that.

12. Discussion: Please present and organize the Discussion as follows: a short, clear summary of the article's findings; what the study adds to existing research and where and why the results may differ from previous research; strengths and limitations of the study; implications and next steps for research, clinical practice, and/or public policy; one-paragraph conclusion. Specifically, please expand the discussion of how your work fits into the context of related and prior literature.

13. References: Please use the "Vancouver" style for reference formatting, and see our website for other reference guidelines https://journals.plos.org/plosmedicine/s/submission-guidelines#loc-references

14. Figures: Please include both titles and legends for each figure. For example, we suggest indicating in the figure legend for Figures 2, 3, 4, 5, and 6 a brief description of what is shown, and the meaning of the points and whiskers. Please also define any abbreviations used within figures.

Comments from the reviewers:

Reviewer #1: See attachment

Michael Dewey

Reviewer #2: Ewoud Schuit, Assistant Professor Clinical Epidemiology, Julius Center for Health Sciences and Primary Care, University Medical Center Utrecht

This systematic review and meta-analysis investigated the impact of visualizing health risk based on medical imaging to patients on their health behaviors and risk factors. The systematic review concerns a relevant topic, and the study has been conducted and written down well.

I do have some comments and suggestions to further improve the paper.

Main comments:

1. I think the current conclusion is formulated too strong. Especially because of the sentence "improve health outcomes". This assumes that healthier behavior results in improved patient outcomes, however, this was not studied in the current review nor was discussed. The paper in general would benefit from linking health behavior and the reduction of risk factor to improved health outcomes in the future, both in the introduction and the discussion of the manuscript. Also, as a reader I would like to know more about how long that change in behavior sticks and whether individuals fall back into their previous 'bad' behavior. Both will have implications for the importance of the outcomes of the current review and meta-analysis.

2. In general, the paper would benefit from additional information on what is considered a minimal clinically relevant difference of the investigated outcomes. Interpretation of SMD's can be difficult, especially when there is no guidance on clinical relevance. 

3. Statistical heterogeneity was assessed based on the chi-square test and i2. I think it would be helpful to further investigate heterogeneity for the outcomes with high heterogeneity. So, can it be explained by certain study characteristics? Did the authors consider for example meta-regression or subgroup analyses to investigate reasons of heterogeneity?

Minor comments:

Abstract

1. The literature search is close to one year ago. An update or at least an assessment of the literature published since 15th July 2020 would be recommended.

2. I would be OK with leaving p-values out of the abstract and only presenting 95% CIs.

3. the authors indicate in the abstract that for "other behavioral outcomes...point estimates favored the intervention but were not statistically significant". Can the authors elaborate on clinical relevance, despite these effects not being statistically significant?

Methods

4. Search. I know some details can be found in the first Cochrane publication, but the current manuscript would benefit from some additional details, e.g.:

a. Did the authors consider searching for ongoing trials, e.g., clinicaltrials.gov? If any trials are ongoing, this may be useful to know and potentially mention in the Discussion section.

b. Did the authors search references of identified studies?

c. Did the authors search grey literature?

d. Were there language restrictions?

5. Appendix 1 presents the detailed MEDLINE search strategy. Please also provide search strategies for the other electronic databases. And in addition to that, provide the number of identified records for each electronic database in separate boxes on the first row of Figure 1.

6. Please explain why nonrandomized trials were not considered.

7. the control group in identified studies either encompasses personal risk information or no such information. Did results of the intervention differ across these control groups, e.g., were seen effects stronger in the latter group of studies?

8. in the data extraction and risk of bias assessment paragraph, consider providing initials of those that performed the screening and extraction.

9. effect sizes for dichotomous outcomes were presented as odds ratios. I was wondering why risk ratios were not considered, provided that included studies were prospective RCTs, meaning absolute risks could be estimated. Risk ratios are easier to interpret than odds ratios.

10. Did the authors consider to study publication bias? If not, please provide an explanation in the manuscript.

Results

11. for medication use, tanning booth use, waist circumference, and FRS, the i2 is high (>70%), which makes it questionable whether data should be pooled in the first place. Did the authors examine where this heterogeneity originated from or tried to account for this (e.g., using meta-regression)?

12. sun protection. Pooled analysis "showed no significant effect". Is and SMD of 0.14 indeed considered "no significant effect", both clinically and statistically? Please change to "statistically significant" if that is what is meant here. A similar comment can be made about blood glucose testing. 

13. consider adding "certainty of evidence" to the results section where results for each of the primary and secondary outcomes are presented. An explanation on GRADE and when the authors considered to downgrade could be added to the methods section.

Discussion

14. There are several additional topics that could be addressed in the discussion section:

a. As indicated earlier, the paper would benefit from linking evidence on health behavior change to change in health outcomes. Also, to know more about how long that change in behavior sticks and whether individuals fall back into their previous 'bad' behavior. Both will have implications for the importance of the outcomes of the current review and meta-analysis.

b. Interventions were generally an hour or less and took place once. How big of an effect would you expect from such an intervention? Could be something to elaborate on a little bit in the discussion.

c. I would be interested in reading more about the implications of the risk of bias and identified heterogeneity across certain outcomes.

Reviewer #3: Thank you for the opportunity to review this manuscript. It is as well-written and important piece of work. I have read the manuscript with great interest and have a few comments for the authors to reflect upon. 

1. P7 1st para last sentence: Did the author exclude studies referred to on the last sentence which commenced with "Studied assessed the intervention effect relative to a comparison group that was either provided…"? The sentence is incomplete. 

2. P7 2nd para on primary outcome: "Primary outcomes were health-related behaviours with the potential to modify communicated health risks". The phrase "to modify communicated health risks" is confusing. Could this be rephrased? 

3. P7 2nd para on secondary outcome: Did the author focus on adverse events and harms, such as anxiety, depression or stress which are related to the intervention only? Did the included studies measure the intervention-related events or harm, or more in general?

4. P8 2nd para: Please clarify the following statement as the use of "combined these data" can lead to misunderstanding. Didn't the author transformed dichotomous data using inverse variance method, so they could finally report the effect sizes as standardised mean differences for all studies homogenously?

"When different studies reported dichotomous and continuous data for the same outcome, we combined these data using the generic inverse variance method."

5. P8 3rd para: The approach the authors used to select the outcome variables (the one with the longest follow-up time, the one which was judged to be most important for risk reduction, and the one which was most stringently measured) lead to heterogeneity of the outcomes measured in different studies. For example, it is inappropriate to use the data points from the longest available follow-up as the outcome would reflect intervention effects at different time points after the intervention. And hence, it is misleading to combine their effects in a meta-analysis. Shouldn't the most common follow-up time, for example, be used and results from all follow-up time are reported separately?

6. P8-P9 on missing data: One can also challenge the approach the authors used to handle missing data. Wouldn't the approach of "assuming that participants with missing outcomes were engaging in the risk increasing behaviour" bias the estimates?

7. P13 on secondary outcome: If possible, add brief detail of the follow-up, especially for the mental health outcomes as the negative effect of intervention on mental health might decay over time. 

8. Results: Some of the secondary outcomes, such as Framingham Risk Score, waist circumference and oral health were significant and had I2 ≥50%. Should the authors consider doing subgroup analysis to explain the high heterogeneity? I would suggest more discussions about the heterogeneity observed in this study. 

9. Results: Many of the studies included have very small sample size. At least the authors can comment on this in the Discussion section. 

10. Conclusion: I think one of the benefits of the widespread and becoming less expensive medical imaging technologies such as carotid ultrasound is its potential to outreach wider population. These technologies could be used for screening of large portion of the population with moderate risk but asymptomatic diseases, serving the purpose of primary prevention for non-communicable diseases. 

Reviewer #4: 

In their systematic review and metanalysis (SR-MA) of visualising health risk with medical imaging the authors show risk-reducing behaviours and reducing risk factors. 

The reviewer has the following comments/questions on the submit: 

The authors are encouraged to formulate a PICO(T) for the SR-MA and especially motivate the great variety of outcome variables. 

Primary and secondary outcomes: The authors should motivate the choice of health risks (primary outcome mandatory for including studies) and secondary risk factors. It appears that for example Framingham Risk Score is more likely to reflect future hard endpoints than for example oral hygiene behaviours. 

The title of the previous Cochrane report "Visual feedback of individuals' medical imaging results for changing health behaviour" appears more distinct and clear compared with the title in the manuscript. 

Abstract; objective: The meaning of "To assess the impact on risk-reducing health behaviours and risk factors of communicating to individuals images representing their bodies derived from medical imaging procedures." is somewhat unclear and should be rephrased. 

Methods, page 7 1st para last sentence: "Studies assessed the intervention effect

relative to a comparison group that was either provided with personalised risk information derived from a non-medical imaging method (e.g. cholesterol test) or from an imaging method but without any visual feedback, or that provided no personalised risk information (e.g. provided only generic health risk information)." Do you mean that these studies were also excluded; please clarify. 

Skin darkening is listed in the results of secondary outcome but is not mentioned in the Methods section. 

Methods, page 8, para 3: The authors state that "If multiple indices of a given behavioural outcome were reported, we used the most stringent and valid measure available (e.g., an objective measure such as biochemically validated smoking cessation)." The authors should describe this in more detail and give more examples. 

Results; table 1: The authors do not consequently specify different kinds of plaque (dental plaques, carotid atherosclerotic plaques); this should be clarified. 

Results, table 1 in the column primary outcome selected for review needs to be revised in more detail. For example: 

Ref 36 Systolic blood pressure, cholesterol (total, TG, LDL), fiber intake, BMI reported in publication but is not described in Table 1 in SR-MA; decrease in fat intake is not an established factor of improved health. 

Ref 38 FRS, cholesterol (total, LDL), blood pressure (syst/diast), medication reported in publication but is missing in the Table 1 in manuscript. 

Ref 40: Electron beam tomography (CT) of what?? FRS (primary endpoint in publication), blood pressure, LDL and more reported in publication but not included in Table 1 in manuscript. 

Ref 42 FRS reported in publication but not in Table 1 in the manus

Ref 43 FRS reported in publication but not in Table 1 in the manus

In the discussion, the relationship between the effect of change in risk/risk factors and health outcomes (hard endpoints such as cancer, CVD, premature death) should be brought up in more detail.

[LINK]

---

## [Decision Letter · Decision Letter 2]

5 Jan 2022

Dear Dr. Hollands,

Thank you very much for re-submitting your manuscript "Visualising health risks with medical imaging for changing recipients’ health behaviours and risk factors: systematic review with meta-analysis" (PMEDICINE-D-21-01143R2) for review by PLOS Medicine.

I have discussed the paper with my colleagues and the academic editor and it was also seen again by the statistical reviewer. I am pleased to say that provided the remaining editorial and production issues are dealt with we are planning to accept the paper for publication in the journal.

[LINK]

We look forward to receiving the revised manuscript by Jan 12 2022 11:59PM.   

Sincerely,

Caitlin Moyer, Ph.D.

Associate Editor 

PLOS Medicine

plosmedicine.org

Requests from Editors:

1. Title: Please capitalize the first word of the subtitle: “Visualising health risks with medical imaging for changing recipients’ health behaviours and risk factors: A systematic review and meta-analysis”

2. Data availability statement: Please clarify the nature of the data extraction files available from the OSF Project page (https://osf.io/bsf5e/). It appears that the protocol is available at this location, but other files are not available. Please clearly explain the location of all data used in the study. If the data are owned by a third party but freely available upon request, please note this and state the owner of the data set and contact information for data requests (web or email address). Note that a study author cannot be the contact person for the data.

3. Abstract: Line 23: It seems as if the CI described for physical activity overlaps with zero. It may be useful to report the exact CI boundary, if different from zero. Otherwise, it may be helpful to clarify the discrepancy between the 95% CIs and the statistical test results in the text.

4. In-text citations: Line 81 (and throughout): Please use square brackets for in-text citations. Where multiple references are noted, please do not include spaces within brackets.

5. Page 29-30: Please remove the Contributors, Funding, Competing Interests, and Data Availability sections from the main text. Please ensure all information is completely and accurately entered into the manuscript submission system. Please move the statement regarding ethical approval to the Methods.

6. References: Please check each reference for the "Vancouver" style for formatting, and see our website for other reference guidelines https://journals.plos.org/plosmedicine/s/submission-guidelines#loc-references

Please check journal title abbreviations (e.g. Lancet, PLoS Med).

7. Table 1: Please be sure all abbreviations are defined in the legend (e.g. CAD).

8. Figure 7: Please make the text size of the figure larger. Please provide a legend/caption for this figure.

9. Appendix 1: We suggest including the search strategies and protocol as two separate supporting information files.

10. PRISMA Checklist: Please revise the checklist to refer to locations within the text only by section and paragraph number. Please do not refer to page numbers. For the final item, please refer to the “Funding” section as the location for this information.

Comments from Reviewers:

Reviewer #1: The authors have addressed my points and the new appendix on outcome selection answers all my concerns. Incidentally I was not regarding multi-level meta-analysis as a panacea and making a choice informed by the scientific and clinical concerns is clearly fully justified.

Michael Dewey

[LINK]

---

## [Editor Report · Decision Letter 3]

19 Jan 2022

Dear Dr Hollands, 

On behalf of my colleagues and the Academic Editor, Sanjay Basu, I am pleased to inform you that we have agreed to publish your manuscript "Visualising health risks with medical imaging for changing recipients’ health behaviours and risk factors: Systematic review with meta-analysis" (PMEDICINE-D-21-01143R3) in PLOS Medicine.

Please also address the following editorial request:

-Title: Please revise the title to: “Visualising health risks with medical imaging for changing recipients’ health behaviours and risk factors: A systematic review with meta-analysis” and please also update the title within the manuscript submission system.

PRESS

Sincerely, 

Caitlin Moyer, Ph.D. 

Associate Editor 

PLOS Medicine